# Min-max approach for comparison of univariate normality tests

**Tanweer Ul Islam** \* 

Department of Economics, School of Social Sciences & Humanities, National University of Sciences and Technology, Islamabad, Pakistan

\* tanweer@s3h.nust.edu.pk

## Abstract

Comparison of normality tests based on absolute or average powers are bound to give ambiguous results, since these statistics critically depend upon the alternative distribution which cannot be specified. A test which is optimal against a certain type of alternatives may perform poorly against other alternative distributions. Thus, an invariant benchmark is proposed in the recent normality literature by computing Neyman-Pearson tests against each alternative distribution. However, the computational cost of this benchmark is significantly high, therefore, this study proposes an alternative approach for computing the benchmark. The proposed *min-max* approach reduces the calculation cost in terms of computing and estimating the Neyman-Pearson tests against each alternative distribution. An extensive simulation study is conducted to evaluate the selected normality tests using the proposed methodology. The proposed *min-max* method produces similar results in comparison with the benchmark based on Neyman-Pearson tests but at a low computational cost.

## 1. Introduction

Normality of the data is the underlying distributional assumption of multitude of statistical procedures and estimation techniques. In both cross-sectional and time series data, assuming the data normality without testing may affect the accuracy of the econometric inference [1]. Statistical inference from regression models applied to time series [2], categorical [3] and count data [4] depends crucially on the assumption of normal errors. The experimental data sets generated in clinical chemistry for the construction of population reference ranges require the assumption of normality [5]. In short, normality assumption of the given data is the key to validate the inferences made from regression models and other statistical procedures. Diagnostic tests for normality are important as Blanca et al. [6], find only 5.5 percent of the 693 real data distributions close to normality while considering skewness and kurtosis together.

Given the importance of the subject, literature has produced a plethora of goodness-of-fit tests to detect departures from normality [7–13]. With the development of several normality tests over the decades, power comparison of these statistics has been given the due consideration in literature in search of the best test thus helping the researchers in the choice of suitable normality test [14–19]. Different characteristics of normal distribution are exploited while developing normality statistics consequently the power of normality tests varies, depending

**Data Availability Statement:** The data generation and replication code file is available by clicking the following link: https://figshare.com/articles/online_resource/Computing_Code_m/14999199.

**Funding:** The author(s) received no specific funding for this work.

**Competing interests:** The authors have declared that no competing interests exist.

upon the nature of non-normality [19]. Thus, one normality statistic may perform well for one alternative distribution and another for another alternative non-normal distribution [18]. Comparison of normality tests via simulations are bound to give ambiguous results, since these statistics critically depend upon the alternative distribution which cannot be specified.

This study rests on the finding that one normality test is optimal against one alternative and another for another alternative distribution [20]. The best test's performance against each alternative distribution provides us the benchmark for comparison of normality tests by using the *max-min* criterion. Maximum deviations of all selected tests from the benchmark is computed and the test with minimum deviation is ranked as best. This method reduces the calculation burden in terms of computing and estimating the Neyman-Pearson test against each alternative distribution for the benchmark as proposed in [18]. Another problem is that the alternative space is infinite dimensional. Since we plan to use numerical methods, we must narrow this space down to something sufficiently small to permit exploration by numerical methods. At the same time, the space should be large enough to provide a good approximation to the full space of alternatives–failing that, it should be large enough to approximate the distributions conventionally used in simulations studies. First and second order departures from normality depend on the skewness and kurtosis of the distribution, we have used 72 alternatives with wider ranges of these parameters. This alternative space includes mixture of uniform distributions, mixture of t-distributions and the distributions used in the literature [14, 16–18, 21].

## 2. Normality tests

This section deals with the background and technical details of the selected normality tests. Each of these tests belongs to a different class of normality tests e.g. ECDF, moments, regression and correlation based tests etc.

In the following literature review, we consider $x_1, x_2, \ldots, x_n$ as a random sample of size $n$. Then $\bar{x}, s^2, \sqrt{b_1} \ and \ b_2$ are the sample mean, variance, skewness and kurtosis respectively, defined as

$$\bar{x} = \frac{\sum_{i=1}^{n} x_i}{n}; \ s^2 = \frac{1}{n-1} \sum_{i=1}^{n} (x_1 - \bar{x})^2 \tag{1}$$

$$\sqrt{b_1} = \frac{m_3}{(s^2)^{3/2}} \ and \ b_2 = \frac{m_4}{(s^2)^2} \tag{2}$$

Where the *rth* central sample moment is defined as

$$m_r = \frac{1}{n} \sum_{i=1}^{n} (x_1 - \bar{x})^r \tag{3}$$

### 2.1. Moment tests

**2.1.1 The Bowman-Shenton $K^2$-test.** Skewness refers to the symmetry of a distribution and Kurtosis refers to the flatness or 'peakedness' of a distribution. These two statistics have been widely used to differentiate between distributions.

The distributions of $\sqrt{b_1}$ and $b_2$ have been approximated by using the Pearson curves. D'Agostino [22] and Anscombe and Glynn [23] have found the normalizing transformations for $\sqrt{b_1}$ and $b_2$ respectively. $Z^2(\sqrt{b_1})$ and $Z^2(b_2)$ denote the resulting approximate standardized normal variables. These can be computed by the following algorithm provided in [24].

Computational Algorithm for $Z(\sqrt{b_1})$:

1. Compute $\sqrt{b_1}$ as defined in (1) above

2. Compute

$$y = \sqrt{b_1} \left( \frac{(n+1)(n+3)}{6(n-2)} \right)^{1/2}$$

Let $k_4$ be the fourth central moment of $\sqrt{b_1}$ then

$$k_4 = \frac{3(n^2 + 27n - 70)(n+1)(n+3)}{(n-2)(n+5)(n+7)(n+9)}$$

$$w^2 = -1 + \{2k_4 - 1\}^{1/2}$$

$$d = 1/\sqrt{\ln w}$$

$$a = \{2/(w^2 - 1)\}^{1/2}$$

*where w, d and a are constants*

3. Compute

$$Z(\sqrt{b_1}) = d\ln(y/a + \{(y/a)^2 + 1\}^{1/2})$$

where $Z(\sqrt{b_1}) \sim N(0,1)$

Computational Algorithm for $Z(b_2)$:

1. Compute $b_2 = m_4/m_2^2$ from the sample data.

2. Compute the mean and variance of $b_2$.

$$E(b_2) = \frac{3(n-1)}{n+1} \qquad \text{and} \qquad var(b_2) = \frac{24n(n-2)(n-3)}{(n+1)^2(n+3)(n+5)}$$

3. Compute the standardized version of $b_2$,

$$x = (b_2 - E(b_2))/\sqrt{var(b_2)}$$

4. Compute the third standardized moment of $b_2$ [24],

$$k_3 = \frac{6(n^2 - 5n + 2)}{(n+7)(n+9)} \sqrt{\frac{6(n+3)(n+5)}{n(n-2)(n-3)}}$$

5. Compute

$$A = 6 + \frac{8}{k_3} \left( \frac{2}{k_3} + \sqrt{\left( 1 + \frac{4}{k_3^2} \right)} \right)$$

6. Compute

$$Z(b_2) = \left( \left( 1 - \frac{2}{9A} \right) - \left[ \frac{1 - 2/A}{1 + x\sqrt{2/(A-4)}} \right]^{1/3} \right) / \sqrt{2/(9A)}$$

D'Agostino and Pearson [8] proposed a test statistic for testing normality that combines $\sqrt{b_1}$ and $b_2$ in the following way:

$$K^2 = Z^2(\sqrt{b_1}) + Z^2(b_2)$$

where $K^2$ is distributed as chi-square with two degrees of freedom. The normality hypothesis is rejected for large values of the test statistic.

**2.1.2 The Jarque-Bera test.** In the field of economics, the most widely used statistics for normality testing is introduced by Jarque and Bera [10, 11]. It is based on the standardized third and fourth moments:

$$JB = n \left( \frac{(\sqrt{b_1})^2}{6} + \frac{(b_2 - 3)^2}{24} \right)$$

where n is the number of observations, $\sqrt{b_1} = m_3/m_2^{3/2}$, $b_2 = m_4/m_2^2$ and $m_i$ is the $i$th central moment of the observations (i.e. $m_i = \sum (x_i - \bar{x})^i / n$). Asymptotically, the JB-statistic is distributed as chi-square with two degrees of freedom. The hypothesis of normality is rejected for large values of the test statistic.

**2.1.3 The Robust Jarque-Bera test.** Gel and Gastwirth [13] introduced robust measures of sample skewness and kurtosis by utilizing a robust measure of dispersion which is less sensitive to outliers, the average absolute deviation from the sample median, and leads to the following robust JB-statistic.

$$RJB = \frac{n}{6} \left( \frac{m_3}{J_n^3} \right)^2 + \frac{n}{64} \left( \frac{m_4 - 3}{J_n^4} \right)^2$$

where

$$J_n = \frac{C}{n} \sum_{i=1}^{n} |X_i - M|, \quad C = \sqrt{\pi/2}$$

where M is the sample median. The $RJB$ statistic asymptotically follows the Chi-square distribution with two degrees of freedom.

**2.1.4 The Bonett-Seier test.** An alternative measure of kurtosis (G-kurtosis) based on Geary's [25] test for normality is defined by Bonett and Seier [12] as

$$w = 13.29((\ln(\sigma) - \ln(\tau))$$

where $\sigma$ *and* $\tau$ is the population standard deviation and mean absolute deviation respectively. The factor of 13.29 is used to scale it up to 3 so that it matches standard measure of kurtosis

and the $E\left[\ln(\hat{\sigma}) - \ln(\hat{\tau})\right] \approx \frac{3}{13.29}$. To test G-kurtosis = 3, Bonett & Seier [12] used the following statistic

$$Z_w = (n+2)^{1/2}(\hat{w} - 3)/3.54$$

$$where \qquad \hat{w} = 13.29(\ln(s) - \ln(t))$$

$$and \;\; s = \sqrt{n^{-1}\sum_{i=1}^{n}(x_i - \bar{x})^2} \quad : \quad t = n^{-1}\sum_{i=1}^{n}|x_i - \bar{x}|$$

The above $Z_w$-statistic is approximately distributed as standard normal.

## 2.2. Distance/ECDF tests

This class of tests deals with the comparison of the empirical cumulative distribution function (ECDF), $F_n\left(x_{(i)}\right) = \frac{i}{n}$, which is estimated based on data with the cumulative distribution function of normal distribution, $Z_i$. Stephens [26] provided versions of the ECDF tests with unknown mean and variance. ECDF tests can be further classified into those involving either the supremum or the square of the discrepancies, $F_n(x_{(i)}) - Z_i$.

$$Z_i = \Phi\left(\frac{x_i - \bar{x}}{s}\right)$$

$$Where \; \bar{x} = \sum x_i/n \; and \; s^2 = \frac{\sum(x_i - \bar{x})^2}{n-1}$$

ECDF tests involving the square of the discrepancies are known as those from the Cramér-von Mises family.

**2.2.1 The Anderson-Darling $A^2$-test.**   Anderson-Darling test is, in fact, a modified form of Cramér-von Mises test. It gives more weight to tails of the distribution than does the Cramér-von Mises test. The computational form of the Anderson-Darling statistic is:

$$A^2 = -\frac{1}{n}\sum_{i=1}^{n}[(2i-1)[\ln(Z_i) + \ln(1 - Z_{n+1-i})]] - n$$

$A^2$-test is the most familiar among all ECDF tests. The asymptotic distribution is known and it was found that the critical values for finite samples quickly converge to their asymptotic values for n ≥ 5.

**2.2.2 The Kolmogorov-Smirnov test.**   In the ECDF class of tests, involving the supremum, a well-known statistic is the Kolmogorov-Smirnov test.

$$KS = \max(D^+, D^-)$$

$$where \; D^+ = max\left(\frac{i}{n} - Z_i\right) \; and \; D^- = max\left(Z_i - \frac{i-1}{n}\right)$$

The null hypothesis of normality is rejected for large values of the test statistic.

**2.2.3 The $Z_a$ and $Z_c$ tests.**   Zhang & Wu [15] proposed two more likelihood ratio statistics of normality testing to the class of EDF tests. The proposed statistics can be defined as follows:

Let $X_{(1)}, X_{(2)}, \ldots, X_{(n)}$ are the ordered statistics from a continuous random variable X with distribution function $F(x)$ to be used for the following hypothesis testing setup.

$$H_0 : F(x) = F_0(x)$$

$$H_a : F(x) \neq F_0(x)$$

where $F_0(x) = \emptyset(x)$ − the cumulative distribution function of the normal distribution.

$$Z_A = - \sum_{i=1}^{n} \left[ \frac{log F_0(X_{(i)})}{n - i + 0.5} + \frac{\log[1 - F_0(X_{(i)})]}{i - 0.5} \right]$$

$$Z_C = \sum_{i=1}^{n} \left[ log \frac{F_0(X_{(i)})^{-1} - 1}{\frac{n - 0.5}{i - 0.75} - 1} \right]^2$$

Null is rejected for large values of the test statistics.

## 2.3. Regression/correlation tests

**2.3.1 The Shapiro-Wilk and Shapiro-Francia tests.** Graphically determining the linearity between the ordered observations $x_{(i)}$ and the expected values of the standard normal ordered statistics, $m_i$ is known as normal probability plotting. The main idea behind these tests is normal probability plotting. Formally, regression or correlation techniques are used to determine the linearity, hence the name of this group of tests.

The Shapiro and Wilk [27] $W$ statistic is defined as the ratio of two estimates of variance of a normal distribution and can be calculated by

$$W = \frac{\left[ \sum_{i=1}^{n} a_i x_{(i)} \right]^2}{\sum_{i=1}^{n} (x_i - \bar{x})^2}$$

The vector of weights can be computed by

$$a' = (a_1, a_2, \ldots, a_n) = m' V^{-1} [(m'V^{-1})(V^{-1}m)]^{-1/2}$$

where $m$ and $V$ are the mean vector and covariance matrix of the ordered statistics of the standard normal distribution [28]. If the distribution of $x_i$ is normal, the $W$-statistic is close to unity otherwise less the unity. The critical values of $W$ are tabulated up to sample sizes of 50. However, Shapiro and Francia [29] noted that as the sample size increases, the ordered observations tends to be independent (i.e. $v_{ij} = 0$ for i≠j). Treating $V$ as an identity matrix, $W$ can be extended for $n$ larger than 50 by

$$W' = \frac{\left[ \sum_{i=n}^{n} m_i x_{(i)} \right]^2}{\sum_{i=1}^{n} (x_i - \bar{x})^2 \sum_{i=1}^{n} m_i^2}$$

Values of $\{m_i\}$ are available in [30] up to sample sizes of 400. However, Weisberg and Bingham [31] suggested the following approximation to compute the values of $\{m_i\}$.

$$m_i \approx \Phi^{-1} \left( \frac{i - \frac{3}{8}}{n + \frac{1}{4}} \right)$$

It was shown that the approximation works even for the small samples as there is no significant difference between the null distributions of $W$ and $W'$ statistics. This simplifies the computation of the test statistics.

**2.3.2 The Chen-Shapiro test.** Chen and Shapiro [32] proposed another competitor of Shapiro-Wilk test based upon the normalized spacing which can be defined as

$$CS = \frac{1}{(n-1)S} \sum_{i=1}^{n} \frac{X_{(i+1)} - X_{(i)}}{H_{i+1} - H_i}$$

where $H_i = \emptyset^{-1} \left[ \frac{i-0.375}{n+0.25} \right]$. $\emptyset^{-1}(.)$ is the inverse of standard normal distribution. Since the authors have shown a close relationship between the Chen-Shapiro ($CS$) and the Shapiro-Wilk ($W$) test, it is therefore expected that the performance of the $CS$ test would be comparable with the $W$ test. The normality hypothesis is rejected for small values of the test statistic.

**2.3.3 The COIN test.** Coin [33] has proposed a normality test especially for the symmetric non-normal alternatives based on polynomial regression. Let $x_{(i)}$ be a vector of ordered observations drawn from a normal population with unknown mean, $\mu$ and variance, $\sigma^2$ then it is possible to write

$$x_{(i)} = \mu + \sigma\alpha_i + \varepsilon_i$$

where $\mu$ and $\sigma$ are the parameters of the best fit line of a normal Q-Q plot and $\varepsilon$ is a vector of errors which are assumed to be homoscedastic. The above two parameters may be estimated by using the Least Square method. Instead of using the above model, COIN proposed the following polynomial model where the vector of ordered observations, $x_{(i)}$ has been replaced by $z_{(i)}$ −a vector of ordered standard normal statistics.

$$z_{(i)} = \beta_1 \alpha_i + \beta_3 \alpha_i^3$$

where $\beta_{i\,(i\,=\,1,3)}$ are the fitting parameters and $\alpha_i$ represents the expected values of standard normal ordered statistics. The estimated value of $\beta_3$ significantly different from zero implies that the sample is drawn from a symmetric non-normal distribution. However, Coin suggests the use of $\beta_3^2$ as a statistic for testing the null hypothesis of normality. Hypothesis of normality is rejected for large values of the test statistic.

**2.3.4 The BCMR test.** Del Barrio, Cuesta-Albertos, Matrán and Rodríguez-Rodríguez [34] proposed the following test statistic for testing normality based on the $L_2$-Wasserstein distance between a sample distribution and the set of normal distributions.

Let $x_1, x_2, \ldots \ldots, x_n$ be a random sample drawn from a distribution with the distribution function F. Let $F_n$ denotes the empirical distribution function, $\emptyset$ the distribution function of the standard normal law and $s^2$ the sample variance.

$$BCMR = 1 - \frac{\left[ \int_0^1 F_n^{-1}(t) \emptyset^{-1}(t) dt \right]^2}{s^2}$$

This statistic is asymptotically equivalent to Shapiro-Wilk and Shapiro Francia statistics [34]. The normality hypothesis is rejected for large values of the statistic.

## 2.4. Other tests

**2.4.1 The Gel-Miao-Gastwirth test.** Recently, Gel and Gastwirth [13] have contributed to the literature of directed tests of normality by proposing a statistic which focuses on detecting heavy tails and outliers of symmetric distributions. The test statistic is simply the ratio of standard deviation to the robust measure of dispersion which should tend to unity under

normality of data.

$$R = \frac{S}{J_n} \qquad J_n = \frac{\sqrt{\pi/2}}{n} \sum_{i=1}^{n} |X_i - M|$$

where M is the median of the sample data. Normality hypothesis is rejected for large values of the statistic. However, the statistic, $\sqrt{n}(R-1)$ is asymptotically distributed as normal with zero mean and standard deviation equal to $\left(\frac{\pi}{2} - 1.5\right)$. The applications of this test can be extended to light tailed distributions as well by using two-sided test for rejecting the null hypothesis of normality.

## 3. Alternative distributions

As already stated, to permit exploration by numerical methods we must narrow down the infinite dimensional alternative space to a space large enough to approximate the distributions conventionally used in simulations studies. We have used 72 alternative distributions with wide ranges of skewness and kurtosis as the first and second order departures from normality depend on these parameters. The simulation study considers the distributions used in the literature (Table 1), mixture of uniform distributions, and mixture of t-distributions (Table 2). The alternate space includes all kind of (a)symmetric, short- and long-tailed distributions.

The mixtures of t- & uniform distributions are generated by the following rules.

$$\lambda.t(v_1; \ \eta_1) + (1 - \lambda).t(v_2; \ \eta_2) \tag{4}$$

$$\lambda.U(a_1, b_1) + (1 - \lambda).U(a_2, b_2) \tag{5}$$

where, $v_i$ & $\eta_i$ ($i$ = 1,2) are the degrees of freedom and the means of the respective t-distributions and $a_i$ & $b_i$ ($i$ = 1,2) are the bounds of uniform distributions.

**Table 1. General distributions.**

| Sr. No. | Distribution | Skewness | Kurtosis | Sr. No. | Distribution | Skewness | Kurtosis |
|---|---|---|---|---|---|---|---|
| 1 | Beta(4,0.5) | -1.79 | 6.35 | 18 | Weibull (2,3.4) | 0.05 | 2.71 |
| 2 | Beta(5,1) | -1.18 | 4.20 | 19 | Gamma(100,1) | 0.20 | 3.06 |
| 3 | Beta(2,1) | -0.57 | 2.40 | 20 | Gamma(15,1) | 0.52 | 3.40 |
| 4 | Weibull (3,4) | -0.09 | 2.75 | 21 | Beta (2,5) | 0.60 | 2.88 |
| 5 | Beta(0.5,0.5) | 0.00 | 1.50 | 22 | Weibull (1,2) | 0.63 | 3.25 |
| 6 | Beta(1,1) | 0.00 | 1.80 | 23 | Gamma(9,1) | 0.67 | 3.67 |
| 7 | Tukey(2) | 0.00 | 1.80 | 24 | Chi2 (10) | 0.89 | 4.20 |
| 8 | Tukey(0.5) | 0.00 | 2.08 | 25 | Gamma (5,1) | 0.89 | 4.20 |
| 9 | Beta (2,2) | 0.00 | 2.14 | 26 | Gumbel (1,2) | 1.14 | 5.40 |
| 10 | Tukey(5) | 0.00 | 2.90 | 27 | Chi2 (4) | 1.14 | 6.00 |
| 11 | Tukey(0.14) | 0.00 | 2.97 | 28 | Gamma (3,2) | 1.15 | 5.00 |
| 12 | t(10) | 0.00 | 4.00 | 29 | Gamma (2,2) | 1.41 | 6.00 |
| 13 | Logistic (0,2) | 0.00 | 4.20 | 30 | Chi2 (2) | 2.00 | 9.00 |
| 14 | Tukey (10) | 0.00 | 5.38 | 31 | Weibull (0.5,1) | 2.00 | 9.00 |
| 15 | Laplace (0,1) | 0.00 | 6.00 | 32 | Chi 2 (1) | 2.83 | 15.00 |
| 16 | t(4) | 0.00 | -- | 33 | LN (0,1) | 6.18 | 113.90 |
| 17 | t(2) | 0.00 | -- | 34 | Cauchy (0,1) | -- | -- |

**Table 2. Mixture of uniform & t-distributions.**

| | Uniform Distributions | | | | | t-distributions | | | | |
|---|---|---|---|---|---|---|---|---|---|---|
| Sr. | $U_1$ | $U_2$ | $\lambda$ | Skew | Kurt | $t_1$ | $t_2$ | $\lambda$ | Skew | Kurt |
| 1 | (-8, -2) | (0, 4) | 0.2 | -1.31 | 3.66 | (10, 3) | (5, 50) | 0.5 | 0.00 | 1.01 |
| 2 | (1, 2) | (3, 5) | 0.1 | -1.18 | 4.11 | (100, -4) | (75, 4) | 0.5 | 0.00 | 1.23 |
| 3 | (-2, 1) | (0, 2) | 0.1 | -0.94 | 4.42 | (100, 4) | (75, 6) | 0.5 | 0.00 | 2.53 |
| 4 | (-8, -2) | (0, 4) | 0.3 | -0.86 | 2.37 | (8, 5) | (10, 3) | 0.5 | 0.04 | 3.02 |
| 5 | (-2, 1) | (0, 2) | 0.3 | -0.75 | 3.01 | (5, 2) | (7, 4) | 0.7 | 0.09 | 4.95 |
| 6 | (1, 2) | (3, 5) | 0.3 | -0.47 | 1.79 | (10, 5) | (5, 7) | 0.5 | 0.16 | 4.20 |
| 7 | (-2, 1) | (0, 2) | 0.5 | -0.40 | 2.26 | (100, 4) | (75, 6) | 0.7 | 0.27 | 2.77 |
| 8 | (-2, -4) | (-1, 4) | 0.37 | 0.00 | 1.62 | (8, 5) | (10, 3) | 0.1 | 0.30 | 3.95 |
| 9 | (-2, -1) | (-1, 2) | 0.25 | 0.00 | 1.80 | (8, 5) | (10, 3) | 0.2 | 0.32 | 3.57 |
| 10 | (2, 5) | (4, 10) | 0.229 | 0.00 | 1.93 | (100, -4) | (75, 4) | 0.9 | 0.36 | 3.39 |
| 11 | (2, 5) | (4, 8) | 0.36 | 0.00 | 2.05 | (10, 5) | (5, 7) | 0.9 | 0.38 | 4.65 |
| 12 | (-10, 0) | (-5, 8) | 0.296 | 0.00 | 2.13 | (100, -4) | (75, 4) | 0.7 | 0.78 | 1.93 |
| 13 | (-2, 1) | (0, 2) | 0.1 | 0.09 | 2.07 | (5, 10) | (7, 25) | 0.7 | 0.82 | 1.83 |
| 14 | (1, 2) | (3, 5) | 0.5 | 0.20 | 1.44 | (10, 3) | (5, 50) | 0.7 | 0.87 | 1.77 |
| 15 | (1, 2) | (3, 5) | 0.7 | 0.96 | 2.38 | (8, 0) | (12, 5) | 0.9 | 1.31 | 5.11 |
| 16 | (-8, -2) | (0, 4) | 0.9 | 1.11 | 4.05 | (8, 0) | (12, 5) | 0.95 | 1.32 | 6.63 |
| 17 | (1, 2) | (3, 5) | 0.9 | 2.35 | 8.08 | (8, -1) | (12, 5) | 0.9 | 1.58 | 5.60 |
| 18 | (1, 2) | (3, 5) | 0.95 | 3.08 | 14.16 | (8, -10) | (12, 5) | 0.9 | 1.78 | 6.02 |
| 19 | -- | -- | -- | -- | -- | (5, 10) | (7, 25) | 0.9 | 2.36 | 7.35 |
| 20 | -- | -- | -- | -- | -- | (10, 3) | (5, 50) | 0.9 | 2.64 | 8.06 |

## 4. Simulation study

An extensive simulation study is conducted in the following to estimate the size and power of the selected normality tests. First, exact critical values are obtained for samples of size 25, 50 & 75 from normal distribution, at 0.05 percent level of significance on the basis of 100, 000 Monte Carlo simulations in MATLAB R2013a. Second, powers of the fourteen normality statistics are computed against general distributions, mixture of uniform and t- distributions.

As stated earlier, no normality test can be uniformly most powerful against all alternative distributions, one test is optimal for one alternative and another is optional for another alternative. The trajectory of maximum power obtained by any test against each alternative distribution provides us the benchmark against which all tests can be compared. Deviations for each test are computed with reference to the benchmark.

Any function, $T(x)$, which takes values {0. 1} is called hypothesis test. The size of the test is defined as

$$\alpha = size(T) = Sup\ P(T = 1/\varphi)$$

where $\varphi$ belongs to null space, $\Phi$. The power of the test is the probability of not committing type-II error i.e.,

$$\pi(T, \varphi) = 1 - \{P(T = 0/\varphi \epsilon \Phi_a)\}$$

For any test, maximum achievable power for a given alternative is defined as

$$Max\ \pi(T, \varphi) = Sup\{P(T = 1/\varphi \epsilon \Phi_a)\}$$

For different values of $\varphi$, we get different optimal tests statistics. The locus of the powers of these statistics provides us the benchmark. Following loss function is computed to evaluate each normality test in terms of its deviation from the benchmark.

$$Deviation = Max\ \pi(T, \varphi) - \pi(T, \varphi)$$

A test with minimum loss or deviation is defined as the best test. The most stringent test will have zero percent loss or deviation from the benchmark. This allows us to rank the normality tests in a unique manner.

## 5. Results & discussion

Normality tests are evaluated against 72 alternative distributions including mixture of uniform distributions (18 distributions), mixture of t-distributions (20 distributions) and the distributions used in the literature (34 distributions) with wide ranges of skewness and kurtosis. The alternate space includes all kind of (a)symmetric, short (long)-tailed distributions. The most stringent test is the one with minimum deviation from the benchmark.

While evaluating the losses or deviations of normality statistics against the selected alternative space, *CS* test outperforms the remaining tests for small (n = 25) and medium (n = 50) sample sizes at 5 percent level of significance with 12.3 & 17.3 percent respective deviations from the benchmark (Table 3). These results corroborate with the findings in [17, 18]. Shapiro-Wilk's *W*-test is the first ranked statistic for large sample size (n = 75) with 27.4 percent deviation closely followed by *CS*, $Z_c$, & $Z_a$ statistics. For third rank, this study recommends *BCMR*, $A^2$, & $Z_a$, tests for small and *BCMR* for medium and large sample sizes. The *JB* and *RJB* tests perform poorly with more than 90 percent losses for all sample sizes which is in line with the findings in [18].

These results clearly indicate that the *min-max* strategy adopted in this study produces similar results as achieved with Neyman-Pearson benchmark in Islam (2017). Furthermore, the computational cost reduces significantly.

It is interesting to note that symmetric short- and long-tailed alternative distributions are the worst alternatives for both the top ranked statistics, *CS* and *W*, in terms of maximum deviations from the benchmark for all sample sizes (Table 4). The *W*-test also outperforms the

**Table 3. Ranking of tests at $\alpha$ = 0.05.**

| n = 25 | | | n = 50 | | | n = 75 | | |
|---|---|---|---|---|---|---|---|---|
| **Test** | **Rank** | **Loss** | **Test** | **Rank** | **Loss** | **Test** | **Rank** | **Loss** |
| CS | 1 | 12.3% | CS | 1 | 17.3% | W | 1 | 27.4% |
| W | 2 | 15.5% | W | 2 | 20.5% | $Z_c$ | 2 | 30.0% |
| BCMR | 3 | 20.6% | $Z_c$ | 2 | 20.5% | CS | 2 | 30.6% |
| $A^2$ | 3 | 21.5% | $Z_a$ | 2 | 21.0% | $Z_a$ | 2 | 32.5% |
| $Z_a$ | 3 | 22.0% | BCMR | 3 | 24.3% | BCMR | 3 | 34.0% |
| $Z_c$ | 4 | 24.8% | $A^2$ | 4 | 33.4% | $A^2$ | 4 | 38.5% |
| $W'$ | 5 | 32.8% | $W'$ | 5 | 44.7% | $W'$ | 5 | 49.8% |
| $K^2$ | 6 | 59.9% | $K^2$ | 6 | 67.8% | R | 6 | 85.2% |
| R | 7 | 76.5% | R | 7 | 75.7% | JB | 6 | 85.9% |
| $Z_w$ | 7 | 78.3% | $Z_w$ | 8 | 88.3% | $Z_w$ | 7 | 88.3% |
| COIN | 8 | 85.5% | JB | 8 | 89.9% | $K^2$ | 8 | 96.3% |
| KS | 9 | 95.4% | COIN | 9 | 96.2% | KS | 8 | 97.1% |
| JB | 9 | 97.3% | KS | 9 | 96.8% | COIN | 8 | 97.4% |
| RJB | 9 | 98.2% | RJB | 10 | 100.0% | RJB | 9 | 100.0% |

**Table 4. Worst alternatives for CS & W test.**

| Distribution | Skewness | Kurtosis | CS-Loss | W-Loss |
|---|---|---|---|---|
| | | n = 25 | | |
| U(-2,-1)*0.25+t(-1,2)*0.75 | 0.00 | 1.80 | 11.3% | 15.0% |
| Tukey(2) | 0.00 | 1.80 | 11.5% | 15.5% |
| Beta(1,1) | 0.00 | 1.80 | 11.6% | 15.5% |
| Laplace (0,1) | 0.00 | 6.00 | 12.3% | 10.8% |
| | | n = 50 | | |
| U(-2,-1)*0.25+t(-1,2)*0.75 | 0.00 | 1.80 | 10.2% | 16.3% |
| Tukey(2) | 0.00 | 1.80 | 10.0% | 16.1% |
| Beta(1,1) | 0.00 | 1.80 | 10.1% | 16.4% |
| U(2,5)*0.229+t(4,10)*0.771 | 0.00 | 1.93 | 10.2% | 17.4% |
| Beta (2,2) | 0.00 | 2.14 | 12.2% | 16.4% |
| Tukey(5) | 0.00 | 2.90 | 12.9% | 11.8% |
| t(4) | 0.00 | -- | 13.0% | 9.9% |
| Tukey(0.5) | 0.00 | 2.08 | 15.2% | 20.5% |
| Laplace (0,1) | 0.00 | 6.00 | 17.3% | 13.5% |
| | | n = 75 | | |
| Logistic (0,2) | 0.00 | 4.20 | 13.2% | 10.0% |
| t(4) | 0.00 | -- | 13.7% | 10.1% |
| Tukey(5) | 0.00 | 2.90 | 15.2% | 14.2% |
| Laplace (0,1) | 0.00 | 6.00 | 17.2% | 12.5% |
| Beta (2,2) | 0.00 | 2.14 | 19.3% | 25.5% |
| Tukey(0.5) | 0.00 | 2.08 | 19.8% | 27.4% |
| U(-10,0)*0.296+U(-5,8)*0.704 | 0.00 | 2.13 | 30.6% | 7.60% |

$A^2$-test with significant lesser deviations from the benchmark which is in line with the findings in [35].

These two statistics along with the $W'$, BCMR & COIN tests belongs to the 'regression & correlation' class of normality tests. The worst alternatives for the rest of the members of this class are test dependent e.g., the worst alternatives for the COIN tests are skewed and the near normal distributions. On balance, when considering the performance of the regression and correlation-based group of normality statistics, CS is the best test (rank#1) for small and medium sample size closely followed by the W test at rank two position. For large samples, the W test outperforms the CS statistics by a margin of 3.2 percent and occupies the first rank position. These two statistics are closely followed by the BCMR test which is placed at rank three position for all sample sizes. The Shapiro-Francia's test ($W'$) shows consistent performance by occupying the fifth position for all sample sizes with maximum deviations ranging from 33 to 50 percent.

Moment based JB & RJB tests perform poorly against the short-tailed symmetric and slightly skewed alternatives (Figs 1 and 2). It is pertinent to mention that the performance of JB & RJB is same at medium sample size (n = 50). Other moment based tests under consideration in this study are $K^2$ & $Z_w$. The $K^2$ statistic is ranked at 6 for small and medium and 8 for large samples sizes with deviations ranging from 60 to 96 percent. The $Z_w$ test is ranked at 7th position for small and large and at 8th for medium sample sizes with maximum deviations from the benchmark range from 78 to 88 percent.

Among the normality tests based on empirical cumulative distribution function (ECDF), $A^2$, $Z_a$ and $Z_c$ occupy the third and fourth rank respectively for small samples. The normality

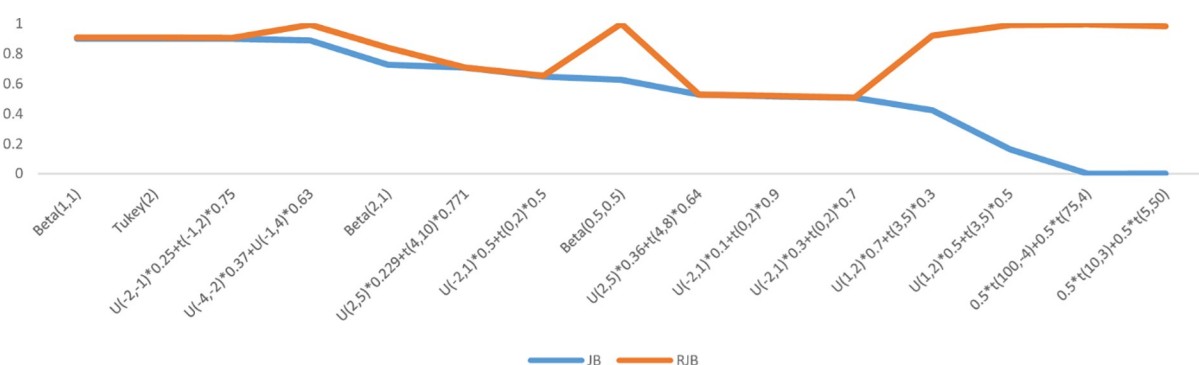

**Fig 1. Worst alternatives for JB & RJB (n = 25) in terms of deviations.**

tests proposed by Zhang & Wu (2005), $Z_a$ & $Z_c$, performed well by occupying the second rank with respective maximum losses of 21.0 & 20.5 percent for small and 32.5 & 30.0 percent for large sample sizes. The Anderson-Darling's statistic ($A^2$) is ranked at fourth position for medium and large sample sizes with 33.4 and 38.5 percent losses. Our results corroborate with the findings in Zhang & Wu (2005). Among the ECDF group of normality tests, the *KS* and *COIN* tests did not perform well with above 85 percent deviations from the benchmark for all sample sizes. The COIN test has slight edge to the *KS* statistic in small sample sizes. For medium and large sample sizes, both share the same ranks with losses above 95 percent.

The *R* statistic introduced by Gel and Gastwirth [13] for symmetric distributions occupies rank 7 for small and medium sample sizes and rank 6 for large sample sizes. Range of the deviation from the benchmark is 76–85 percent when evaluated against the entire class of alternatives. The worst distributions for the *R* test belong to asymmetric alternative space for the obvious reasons. Interestingly, the *R* test occupies sixth and fifth ranks for small and medium to large sample sizes respectively when evaluated against the symmetric alternatives (Table 5). The worst distributions for the *R* test belongs to symmetric short-tailed alternative space (Fig 3) for all sample sizes. Therefore, *R* test is not recommended for symmetric short-tailed alternatives. The COIN test perform relatively much better than the *R* test which is inline with the findings in [17, 33].

While considering symmetric alternative space, the *CS & COIN* are the best options for testing normality for small to medium sample sizes, the Shapiro-Wilk's *W* test is recommended for large sample sizes (Table 5). The *W*-test occupies second rank for small and medium

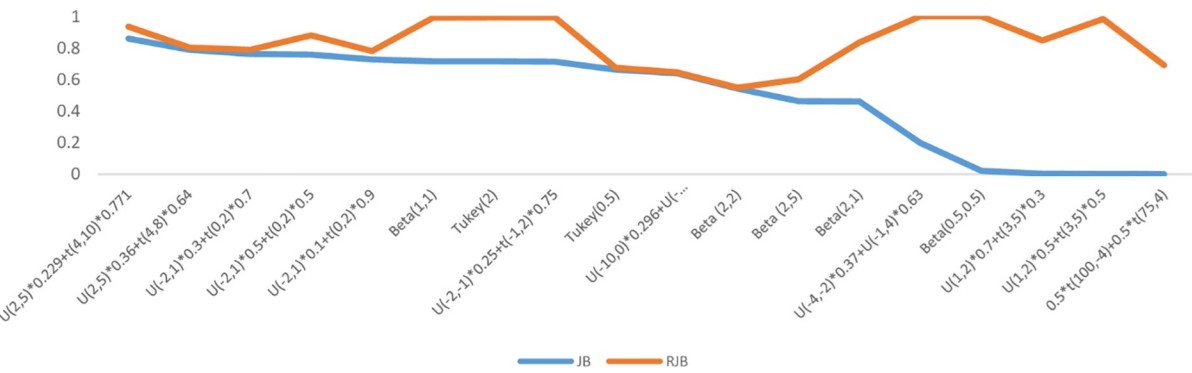

**Fig 2. Worst alternatives for JB & RJB (n = 75) in terms of deviations.**

**Table 5. Ranking of tests against symmetric alternatives.**

| n = 25 | | | n = 50 | | | n = 75 | | |
|---|---|---|---|---|---|---|---|---|
| Test | Rank | Loss | Test | Rank | Loss | Test | Rank | Loss |
| CS | 1 | 12.3% | CS | 1 | 17.3% | W | 1 | 27.4% |
| COIN | 1 | 14.1% | COIN | 1 | 19.0% | $Z_c$ | 2 | 30.0% |
| W | 2 | 15.5% | W | 2 | 20.5% | CS | 2 | 30.6% |
| BCMR | 3 | 20.6% | $Z_c$ | 2 | 20.5% | $Z_a$ | 3 | 32.5% |
| $A^2$ | 3 | 21.5% | $Z_a$ | 2 | 21.0% | COIN | 3 | 33.9% |
| $Z_a$ | 3 | 22.0% | BCMR | 3 | 24.3% | BCMR | 3 | 34.0% |
| $Z_w$ | 4 | 23.1% | $Z_w$ | 4 | 26.5% | $A^2$ | 4 | 38.5% |
| $Z_c$ | 4 | 24.8% | R | 5 | 28.7% | $Z_w$ | 4 | 40.0% |
| $W'$ | 5 | 32.8% | $A^2$ | 6 | 33.4% | R | 5 | 45.8% |
| R | 6 | 36.1% | $W'$ | 7 | 44.7% | $W'$ | 6 | 49.8% |
| $K^2$ | 7 | 43.9% | $K^2$ | 8 | 67.8% | KS | 7 | 85.3% |
| KS | 8 | 91.6% | JB | 9 | 89.9% | JB | 7 | 85.9% |
| JB | 9 | 97.2% | KS | 9 | 90.9% | $K^2$ | 8 | 96.3% |
| RJB | 9 | 98.2% | RJB | 10 | 100.0% | RJB | 9 | 100.0% |

sample sizes. The moment-based *JB & RJB* tests performed poorly against the symmetric class of alternatives as well. The worst distributions for these statistics belongs to symmetric and short-tailed class of alterntaives. These results corroborate with the findings in [16, 35, 36]. The $Z_w$ test perfoms relatively well among the moment-based normality tests and occupies fourth rank for all sample sizes with maximum deviation from the benchmark ranging between 23–40 percent. The $K^2$ test is ranked at 7th & 8th positions for small and medium to large samples respectively with losses range of 44–96 percent.

When considering the regression and correlation based group of normality tests, the *CS, COIN, W* & BCMR are the best options against the symmetric alternatives and occupy top three ranks in the table. Romão et al. [17], recommend the *CS & W* statistics for asymmetric group of alternatives by comparing the absolute powers. However, when these statistics are evaluated against a benchmark instead of absolute powers, these statistics turn out to be best for symmetric alternative distributions as well. The Shapiro-Francia's ($W'$) test does not perform well against symmetric alternaives and occupies ranks 5, 7 & 6 for small, medium and large smaple sizes respectively.

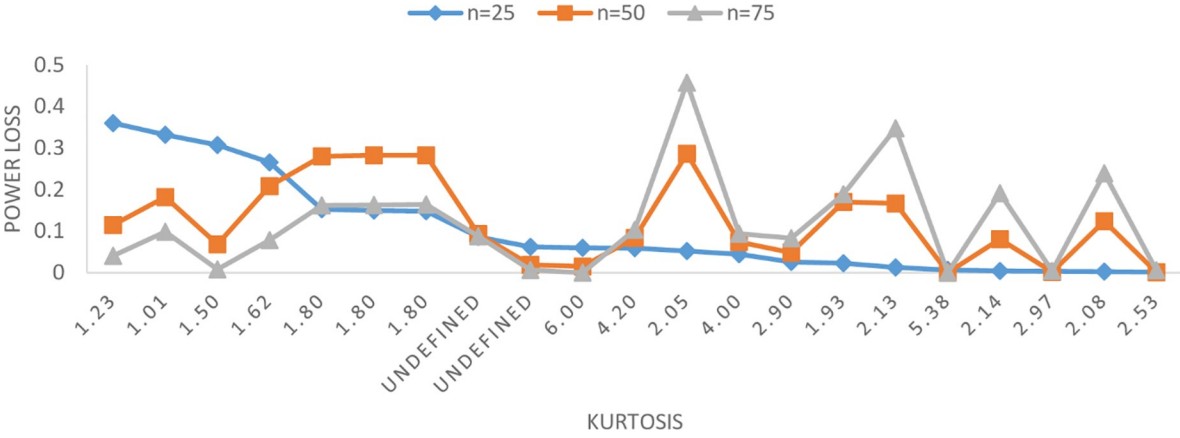

**Fig 3. Worst symmetric alternatives for R test.**

**Table 6. Ranking of tests against asymmetric alternatives.**

| | n = 25 | | | n = 50 | | | n = 75 | |
|---|---|---|---|---|---|---|---|---|
| Test | Rank | Loss | Test | Rank | Loss | Test | Rank | Loss |
| W | 1 | 5.0% | CS | 1 | 6.9% | CS | 1 | 9.0% |
| $Z_c$ | 1 | 5.3% | W | 2 | 9.5% | W | 1 | 10.2% |
| CS | 1 | 5.4% | $Z_c$ | 2 | 10.2% | $Z_c$ | 1 | 10.3% |
| BCMR | 2 | 7.7% | $Z_a$ | 3 | 12.2% | $Z_a$ | 1 | 10.5% |
| $Z_a$ | 2 | 9.0% | BCMR | 4 | 15.3% | BCMR | 2 | 13.6% |
| $W'$ | 3 | 15.8% | $A^2$ | 5 | 30.7% | $A^2$ | 3 | 26.5% |
| $A^2$ | 4 | 21.0% | $W'$ | 5 | 31.4% | $W'$ | 4 | 33.4% |
| KS | 5 | 49.0% | $K^2$ | 6 | 64.9% | $K^2$ | 5 | 50.6% |
| $K^2$ | 6 | 59.9% | KS | 6 | 65.5% | KS | 6 | 63.4% |
| R | 7 | 76.5% | JB | 7 | 72.6% | JB | 7 | 76.2% |
| $Z_w$ | 7 | 78.3% | R | 8 | 75.7% | R | 8 | 85.2% |
| COIN | 8 | 85.5% | $Z_w$ | 9 | 88.3% | $Z_w$ | 9 | 88.3% |
| JB | 9 | 97.3% | COIN | 10 | 96.2% | COIN | 10 | 97.4% |
| RJB | 9 | 97.7% | RJB | 11 | 99.0% | RJB | 11 | 98.4% |

Among the ECDF class of normality tests, $A^2$ & $Z_a$ occupy the third rank for small samples with 21.5 & 22.0 percent deviations from the benchmark. The $Z_c$ & $Z_a$ are recommended for medium and large sample sizes as Anderson-Darling's statistic ($A^2$) occupies sixth and fourth positions for medium and large sample sizes respectively. In terms of maximum deviatoins, the $Z_c$ has slight edge to $Z_a$ test for medium and large sample sizes which does not corroborate with the findings in [15]. The KS test does not perform well against the symmetric alternatives with more than 85 percent losses for all sample sizes.

When the selected normality tests are evaluated against the asymmetric class of alternatives, W, $Z_c$, & CS tests occupy the rank one position for small, CS for medium, and CS, W, $Z_c$ & $Z_a$ for large sample sizes (Table 6). On balance, the CS and W tests from regression and correlation based group of normality tests is recommended for all sample sizes whereas the COIN test did not perform well with very high range of deviations from the benchmark when the alternative distribution is drawn from the asymetric distributional space due to obvious reasons. These findings are corroborate with the findings in [17, 18, 35].

Among the ECDF class of tests, the $Z_c$ is ranked as number one statistic for small & large sample sizes and number two for medium sample size against the selected asymmetric distributional space closely followed by $Z_a$ test. Maximum deviations of these tests range from 5 to 12 percent. TheAnderson-Darling test, $A^2$, is placed at fourth, fifth, and third positions for small, medium, and large sample sizes, respectively with a range of 21–31 percent maximum deviations from the benchmark. Moment-based tests did not perform well with more than 50 percent maximum deviations from the benchmark for all sample sizes against the asymmetric distributional space.

There is no significant difference between the performances of BCMR and W tests of normality in terms of discriminating the long-tailed distributions ($\beta_2 > 3$). Both the statistics share first rank when evaluated against the selected class of heavy tailed distributional space (Table 7) closely followed by the CS test. The Shapiro-Francia's $W'$ test performed well for small samples and occupies third rank with power loss of 15.1 percent however, from regression and correlation class, the COIN test peforms poorly and occupies the last rank with more than 85 percent power losses at all sample sizes. On balance, in terms of maximum deviations from the benchmark, moment-based normality tests do not perform well (Table 7).

**Table 7. Ranking of the tests against long-tailed alternatives.**

| n = 25 | | | n = 50 | | | n = 75 | | |
|---|---|---|---|---|---|---|---|---|
| Test | Rank | Loss | Test | Rank | Loss | Test | Rank | Loss |
| BCMR | 1 | 8.9% | W | 1 | 13.5% | W | 1 | 12.5% |
| W | 1 | 10.8% | BCMR | 1 | 15.3% | BCMR | 1 | 13.6% |
| CS | 2 | 12.3% | CS | 2 | 17.3% | CS | 2 | 17.2% |
| $W'$ | 3 | 15.1% | $A^2$ | 2 | 17.3% | $Z_c$ | 3 | 22.0% |
| $Z_a$ | 3 | 15.4% | $Z_a$ | 3 | 19.6% | $Z_a$ | 3 | 22.9% |
| $A^2$ | 3 | 16.5% | $Z_c$ | 3 | 19.7% | $A^2$ | 4 | 26.5% |
| $Z_c$ | 4 | 24.8% | $W'$ | 4 | 31.4% | $W'$ | 5 | 33.4% |
| JB | 5 | 37.8% | $K^2$ | 4 | 32.3% | $K^2$ | 5 | 35.7% |
| $K^2$ | 6 | 43.9% | KS | 5 | 34.9% | KS | 6 | 43.0% |
| RJB | 6 | 44.4% | JB | 6 | 51.5% | R | 7 | 70.6% |
| KS | 7 | 49.0% | RJB | 6 | 51.8% | JB | 8 | 76.2% |
| R | 8 | 66.4% | R | 7 | 75.7% | RJB | 9 | 78.7% |
| $Z_w$ | 9 | 78.3% | $Z_w$ | 8 | 88.3% | $Z_w$ | 10 | 88.3% |
| COIN | 10 | 85.5% | COIN | 9 | 96.2% | COIN | 11 | 97.4% |

However, these statistics really performed well when evaluated against the symmetric long-tailed distributions with clear dominance of the *RJB* test (Table 8). These results are in line with the findings in [18, 33, 36]. Overall, the *JB & RJB* tests perform well against the long-tailed distributional space except for the alternatives listed in Table 9. Among the ECDF class of normality tests, all the statistics except for *KS* performed well and are listed among the top four tests for all sample sizes. The *R* test when evaluated against the thick-tailed alternatives do not perform well and power deviations vary from 66–76 percent (Table 7).

For distributions from the short-tailed alternative space ($\beta_2 < 3$), we recommend *CS* & $Z_c$ for small, *CS* for medium and *W* test for large sample sizes (Table 10). Romão et al. [17], also recommends the use of *CS* & *W* tests for small and large sample sizes. The *W*-test is also ranked second for small and medium sample sizes with respective maximum deviations of 15.5 & 20.5 percent from the benchmark. Performance of the *W* test is much better than the *KS* test irrespective of the fact that the alternative belongs to short- or long-tailed distributional space which corroborates with the findings in [18, 35]. Both the $Z_a$ & $Z_c$ statistics are among the top three positions with $Z_c$ having a slight edge to $Z_a$ against the short-tailed alternatives which is in line with the findings in [18]. Anderson-Darling test statistic ($A^2$) also performs well and occupies third & fourth ranks for small and medium & large samples, respectively. Based on the maximum deviations from the benchmark, *BCMR* test is placed at rank three with respective power losses of 20.6, 24.3, & 34.0 percent. Among the correlation and regression-based normality tests, the *COIN* test could not perform well when evaluated separately

**Table 8. Powers of moment-based tests against symmetric long-tailed alternatives.**

| Distribution | Skewness | Kurtosis | $K^2$ | JB | RJB | $Z_w$ | Best Test |
|---|---|---|---|---|---|---|---|
| t(10) | 0.00 | 4.00 | 0.23 | 0.26 | 0.27 | 0.17 | 0.27 |
| Logistic (0,2) | 0.00 | 4.20 | 0.29 | 0.33 | 0.35 | 0.25 | 0.35 |
| Tukey (10) | 0.00 | 5.38 | 0.81 | 0.91 | 1.00 | 1.00 | 1.00 |
| Laplace (0,1) | 0.00 | 6.00 | 0.63 | 0.70 | 0.80 | 0.80 | 0.81 |
| t(4) | 0.00 | . . . | 0.63 | 0.68 | 0.71 | 0.62 | 0.71 |
| t(2) | 0.00 | . . . | 0.95 | 0.96 | 0.98 | 0.97 | 0.98 |

**Table 9. Worse long-tailed alternatives for *JB* & *RJB* (deviations in percentages).**

| Distribution | Skewness | Kurtosis | *JB* | *RJB* |
|---|---|---|---|---|
| | | n = 25 | | |
| Beta(5,1) | -1.18 | 4.20 | 35.20% | 44.38% |
| Tukey (10) | 0.00 | 5.38 | 37.85% | -- |
| U(1,2)*0.9+t(3,5)*0.1 | 2.35 | 8.08 | 24.23% | 32.60% |
| | | n = 50 | | |
| U(-2,1)*0.3+t(0,2)*0.7 | -0.75 | 3.01 | 50.53% | 50.62% |
| U(-2,1)*0.1+t(0,2)*0.9 | -0.94 | 4.42 | 51.49% | 51.81% |
| | | n = 75 | | |
| U(-2,1)*0.3+t(0,2)*0.7 | -0.75 | 3.01 | 76.22% | 78.74% |
| U(-2,1)*0.1+t(0,2)*0.9 | -0.94 | 4.42 | 72.65% | 78.06% |

both for short- and long-tailed alternatives. Performance of the *JB*, *RJB*, $K^2$, & $Z_w$ is not up to the mark with very high-power deviations which corroborates with the findings in [36].

Table 11 presents the top five damaging distributions for each normality test at samples of size 25, 50, & 75. It is evident from the results that ECDF based normality tests suffer more against the symmetric short-tailed and symmetric long-tailed distributions with significant outliers. Symmetric short-tailed and skewed distributions affect the performance of normality tests belong to regression and correlation class. However, the most damaging distributions for the moment based normality tests are specific to individual test in this class. For example, the JB & RJB tests suffer greater power loss against the long-tailed alternatives at small, negatively skewed alternatives at medium and large sample sizes.

## 6. Conclusion

Comparison of normality test without having an invariant benchmark has not been proven fruitful in the normality literature. This study proposes an alternative way to compute the benchmark instead of the Neyman-Pearson test-based benchmark proposed in literature. The proposed benchmark is based on the *min-max* approach which reduces the calculation cost in

**Table 10. Ranking of the tests against short-tailed alternatives.**

| n = 25 | | | n = 50 | | | n = 75 | | |
|---|---|---|---|---|---|---|---|---|
| Test | Rank | Loss | Test | Rank | Loss | Test | Rank | Loss |
| CS | 1 | 11.6% | CS | 1 | 15.2% | W | 1 | 27.4% |
| $Z_c$ | 1 | 12.1% | W | 2 | 20.5% | $Z_c$ | 2 | 30.0% |
| W | 2 | 15.5% | $Z_c$ | 2 | 20.5% | CS | 2 | 30.6% |
| BCMR | 3 | 20.6% | $Z_a$ | 2 | 21.0% | Za | 2 | 32.5% |
| $A^2$ | 3 | 21.5% | BCMR | 3 | 24.3% | BCMR | 3 | 34.0% |
| $Z_a$ | 3 | 22.0% | $A^2$ | 4 | 33.4% | $A^2$ | 4 | 38.5% |
| $W'$ | 4 | 32.8% | $W'$ | 5 | 44.7% | $W'$ | 5 | 49.8% |
| KS | 5 | 48.9% | $K^2$ | 6 | 67.8% | KS | 6 | 63.4% |
| $K^2$ | 6 | 59.9% | $Z_w$ | 7 | 73.1% | $Z_w$ | 7 | 82.2% |
| $Z_w$ | 7 | 62.3% | R | 7 | 74.9% | R | 8 | 85.2% |
| COIN | 8 | 71.4% | KS | 8 | 78.4% | JB | 8 | 85.9% |
| R | 9 | 76.5% | COIN | 8 | 80.0% | COIN | 9 | 92.7% |
| JB | 10 | 97.3% | JB | 9 | 89.9% | $K^2$ | 10 | 96.3% |
| RJB | 10 | 98.2% | RJB | 10 | 100.0% | RJB | 11 | 100.0% |

**Table 11. Top five damaging distributions for normality tests.**

| Sr. | Skew | Kurt | KS | CS | $A^2$ | Za | Zc | $K^2$ | JB | RJB | Zw | W | W' | COIN | BCMR | R |
|---|---|---|---|---|---|---|---|---|---|---|---|---|---|---|---|---|
| D1 | 0.00 | 1.62 |  |  |  |  |  |  | β | βγ |  |  | α |  | α |  |
| D2 | 0.00 | 1.80 | Βγ | α | αβ | α | α |  | β | γ |  | αβ | αβ |  | αβ |  |
| D3 | 0.00 | 1.93 |  |  |  | β | β |  | γ |  |  | β | βγ |  | αβ |  |
| D4 | 0.00 | 2.05 | Γ |  | βγ |  |  |  | γ |  |  |  | βγ | α |  | γ |
| D5 | 0.00 | 2.13 |  | γ |  |  |  |  |  |  |  |  | γ |  |  | γ |
| D6 | -1.31 | 3.66 |  |  |  |  |  |  |  |  |  |  |  |  | βγ |  |
| D8 | -0.94 | 4.42 |  |  |  |  |  |  | γ |  |  |  |  |  |  |  |
| D9 | -0.86 | 2.37 |  |  |  |  |  | α |  |  |  |  |  |  |  | αβ |
| D10 | -0.75 | 3.01 |  |  |  |  |  |  | γ |  |  |  |  |  |  |  |
| D11 | -0.47 | 1.79 |  |  |  |  |  | α |  |  |  |  |  |  |  |  |
| D12 | -0.40 | 2.26 |  |  |  |  |  | βγ | γ |  |  |  |  |  |  | γ |
| D13 | 0.09 | 2.07 | Aβγ |  | αβ |  |  |  |  |  | βγ |  |  | γ |  | βγ |
| D14 | 0.20 | 1.44 |  |  |  |  |  |  | α | αβ |  |  |  |  |  |  |
| D15 | 0.96 | 2.38 |  |  |  |  |  | α | α | α |  |  |  |  |  | αβγ |
| D17 | 2.35 | 8.08 | A |  | α |  |  |  |  |  | αβγ |  |  | αβγ |  | α |
| D19 | 0.00 | 1.50 | Aβ |  |  |  |  |  | α | αβγ |  |  | α |  |  |  |
| D20 | 0.00 | 1.80 | Aβγ | α | αβ | α | α |  | β | γ |  | αβ | αβ |  | αβ |  |
| D21 | 0.00 | 1.80 | Γ | α | αβ | α | α |  | β | γ |  | α | αβ |  | αβ |  |
| D22 | 0.00 | 2.08 |  | βγ | γ | βγ | βγ |  |  |  |  |  | βγ | γ |  | γ |
| D23 | 0.00 | 2.14 |  | βγ | γ | βγ | βγ |  |  |  |  |  | βγ | γ | βγ |  |
| D24 | 0.00 | 2.90 |  | βγ |  | βγ | βγ |  |  |  |  |  | γ |  |  | γ |
| D26 | 0.00 | 4.00 |  |  |  |  |  | α |  |  |  |  |  |  |  |  |
| D28 | 0.00 | 5.38 |  |  |  | α | α | β |  |  |  |  |  |  |  |  |
| D29 | 0.00 | 6.00 | αβγ |  |  | βγ | αβγ |  |  |  |  | αγ |  |  |  |  |
| D30 | 0.00 | .. |  | β |  | γ |  |  |  |  |  |  |  |  |  |  |
| D31 | 0.00 | .. |  | α |  |  | γ |  |  |  |  | α |  |  |  |  |
| D32 | .. | .. |  |  |  |  |  | α |  |  |  |  |  |  |  |  |
| Sr. | Skew | Kurt | KS | CS | $A^2$ | Za | Zc | $K^2$ | JB | RJB | $Z_w$ | W | W' | COIN | BCMR | R |
| D33 | -1.79 | 6.35 |  |  |  |  |  |  |  |  | α |  |  | αβ |  |  |
| D34 | -1.18 | 4.20 |  |  |  |  |  |  |  |  | αβγ |  |  | βγ |  | αβ |
| D35 | -0.57 | 2.40 |  | β |  |  |  |  | βγ | β |  |  |  |  |  | βγ |
| D40 | 0.60 | 2.88 |  |  | γ |  |  | βγ |  |  |  | γ |  |  |  | γ |
| D41 | 0.63 | 3.25 |  |  | γ |  |  |  |  |  |  |  |  |  |  |  |
| D46 | 1.14 | 6.00 |  |  |  |  |  |  |  |  | βγ |  |  | β |  |  |
| D47 | 1.15 | 5.00 |  |  |  |  |  |  |  |  |  |  |  | γ |  |  |
| D48 | 1.41 | 6.00 |  |  |  |  |  |  |  |  | βγ |  |  |  |  |  |
| D49 | 2.00 | 9.00 |  |  |  |  |  |  |  |  | α |  |  | α |  |  |
| D50 | 2.00 | 9.00 |  |  |  |  |  |  |  |  | α |  |  | α |  |  |
| D53 | 0.00 | 1.01 |  |  |  |  |  | βγ | α | α |  |  |  |  |  |  |
| D54 | 0.00 | 1.23 |  |  |  |  |  | γ |  | β |  |  |  |  |  |  |
| D55 | 0.87 | 1.77 |  |  |  |  |  |  |  |  |  |  |  |  |  | α |

Note: α, β, and γ represent the damaging distributions at samples of size 25, 50, and 75 respectively. $D_i$ represent the distributions presented in Tables 1 and 2.

terms of computing and estimating the Neyman-Pearson tests against each alternative from the selected distributional space. The min-max approach is based on the finding that one test is best against one alternative and another for another alternative [20]. Thus, against each

alternative distribution, we get different optimal normality tests. The locus of these statistics provides us the benchmark. *Maximum* deviations from the benchmark are computed for the selected normality statistics. A test with *minimum* loss or deviation is defined as the most stringent test. An extensive simulation study is conducted to rank the selected normality tests against a vast distributional space consisting of mixture of uniform distributions, mixture of t-distributions, and distributions used in literature.

General recommendations derived from the analysis of maximum deviations from the benchmark indicate the most stringent normality test is *CS* for small (n = 25), medium (n = 50), and Shapiro-Wilk's *W*-test for large sample size (n = 75) closely followed by *CS*, $Z_c$, & $Z_a$ statistics against the entire alternative space. While considering symmetric alternative space, the *CS* & *COIN* are the best options for testing normality for small to medium sample sizes, and the Shapiro-Wilk's *W* test for large sample sizes. When the selected normality tests are evaluated against the asymmetric class of alternatives, *W*, $Z_c$, & *CS* tests occupy the rank one position for small, *CS* for medium, and *CS*, *W*, $Z_c$ & $Z_a$ for large sample sizes.

There is no significant difference between the performances of *BCMR* and *W* tests of normality in terms of discriminating the long-tailed distributions ($\beta_2 > 3$). Both the statistics share first rank when evaluated against the selected class of heavy tailed distributional space closely followed by the *CS* test. For distributions from the short-tailed alternative space ($\beta_2 < 3$), we recommend *CS* & $Z_c$ for small, *CS* for medium and *W* test for large sample sizes.

## Author Contributions

**Conceptualization:** Tanweer Ul Islam.

**Formal analysis:** Tanweer Ul Islam.

**Methodology:** Tanweer Ul Islam.

**Software:** Tanweer Ul Islam.

**Writing – original draft:** Tanweer Ul Islam.

**Writing – review & editing:** Tanweer Ul Islam.

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
