## [Decision Letter · Decision Letter 0]

16 Jun 2021

PONE-D-21-14331

An Alternative Approach for Comparison of Univariate Normality Tests

PLOS ONE

Dear Dr. Islam,

Thank you for submitting your manuscript to PLOS ONE. After careful consideration, we feel that it has merit but does not fully meet PLOS ONE’s publication criteria as it currently stands. Therefore, we invite you to submit a revised version of the manuscript that addresses the points raised during the review process.

We look forward to receiving your revised manuscript.

Kind regards,

Saeed Mian Qaisar, Ph.D.

Academic Editor

PLOS ONE

Journal Requirements:

Please review your reference list to ensure that it is complete and correct. If you have cited papers that have been retracted, please include the rationale for doing so in the manuscript text, or remove these references and replace them with relevant current references. Any changes to the reference list should be mentioned in the rebuttal letter that accompanies your revised manuscript. If you need to cite a retracted article, indicate the article’s retracted status in the References list and also include a citation and full reference for the retraction notice

Reviewers' comments:

Reviewer's Responses to Questions

**Comments to the Author**

1. Is the manuscript technically sound, and do the data support the conclusions?

Reviewer #1: Yes

Reviewer #2: Yes

2. Has the statistical analysis been performed appropriately and rigorously? 

Reviewer #1: Yes

Reviewer #2: Yes

3. Have the authors made all data underlying the findings in their manuscript fully available?

Reviewer #1: Yes

Reviewer #2: No

4. Is the manuscript presented in an intelligible fashion and written in standard English?

Reviewer #1: Yes

Reviewer #2: Yes

5. Review Comments to the Author

Reviewer #1: I found the paper very interesting and opening up a new debate on Normality testing. On balance, this is a technically sound piece of research which explores the power properties of normality tests using a diversified alternative distributional space. The data generation, simulation design, and finite sample properties are handled well. The proposed min-max approach definitely reduces the calculation cost in terms of computing power envelope which is the contribution of this paper. However, I have the following minor comments:

1. In section 2, distribution of the RJB test should be mentioned as well.

2. In table 3 & 5, how the ranks are defined for tests with small differences in losses?

3. Fig. 1 & 2 highlight the deviations of JB and RJB test against the worst alternatives at sample of sizes 25 and 75. Do the tests behave alike at sample of size 50? If yes, add this information as footnote or in the main text.

4. Table numbers are not aligned, e.g., there are two tables titled as table 5.

5. It is evident from literature that the JB and RJB tests behave well against long-tailed alternatives. This study should acknowledge the same (as it is evident from this study results as well) and particularly mention that there are only few long-tailed worst alternatives which should be taken care of while testing normality assumption by applying the JB and RJB tests.

Reviewer #2: The paper presents an extensive simulation study on normality tests. It could be useful for researchers in order to select the most appropriate normality test. I have two suggestions to the authors:

1. More details on the simulations should be disclosed. For example, which software do you use? Additionally, in order to facilitate the reproduction of the study you should provide data. If you generate your distributions with Python, R, Matlab, etc, you could include the code as a supplementary material or upload at a repository such as Github.

2. Probably you could complement your results using a classification or a clustering algorithm. In other words, which features of the distribution influences the loss function the most? Do some tests perform better at detecting non normality against a Beta rather than against a Laplace distribution? You could use Self Organizing Maps or K-means or many other techniques in order to produce such representation. IN this way instead of a ranking such as in tables 5,6,7, you could produce a ranking that informs on the different dimensions that affect the normality detection.

6. PLOS authors have the option to publish the peer review history of their article (what does this mean?). If published, this will include your full peer review and any attached files.

Reviewer #1: **Yes: **Asad ul Islam KHAN

Reviewer #2: No

---

## [Author Response · Author response to Decision Letter 0]

26 Jun 2021

Saeed Mian Qaisar, Ph.D.

Academic Editor

PLOS ONE

Manuscript ID: PONE-D-21-14331

Dear Dr. Saeed Mian Qaisar,

Thank you for inviting me to submit a revised draft of our manuscript entitled, "Min-Max Approach for Comparison of Univariate Normality Tests" to PLOS ONE. I also appreciate the time and effort you and each of the reviewers have dedicated to providing insightful feedback on ways to strengthen my paper. Thus, it is with great pleasure that I resubmit my article for further consideration. I have incorporated changes that reflect the detailed suggestions you have graciously provided. I also hope that my edits and the responses I provide below satisfactorily address all the issues and concerns you and the reviewers have noted.

To facilitate your review of my revisions, the following is a point-by-point response to the questions and comments delivered in your letter dated June 16 2021.

Journal Requirements:

Response: I have followed all the style requirements including the file names. 

2. Upon re-submitting your revised manuscript, please upload your study’s minimal underlying data set as either Supporting Information files or to a stable, public repository and include the relevant URLs, DOIs, or accession numbers within your revised cover letter.

Response: This is a simulation based study and all the codes required to generate data or replicate the study are deposited on the Github as suggested by one of the reviewer. The URL of the file is included in the revised cover letter. 

Reviewer # 1 Comments: 

I appreciate the time and energy the anonymous reviewer committed and provided me the valuable feedback for the improvement of the manuscript. 

1. In section 2, distribution of the RJB test should be mentioned as well.

Response: The distribution is mentioned on pg#7, line# 132-33. 

2. In table 3 & 5, how the ranks are defined for tests with small differences in losses?

Response: The ranking changes if the difference in losses is more than 2%. 

3. Fig. 1 & 2 highlight the deviations of JB and RJB test against the worst alternatives at sample of sizes 25 and 75. Do the tests behave alike at sample of size 50? If yes, add this information as footnote or in the main text.

Response: Thanks for highlighting the missing information. Yes, the tests behave alike at sample of size 50 as well and this information is added in main text at pg#21, line# 335-36.

4. Table numbers are not aligned, e.g., there are two tables titled as table 5.

Response: This typo is corrected on pg# 25, line# 397.

5. It is evident from literature that the JB and RJB tests behave well against long-tailed alternatives. This study should acknowledge the same (as it is evident from this study results as well) and particularly mention that there are only few long-tailed worst alternatives which should be taken care of while testing normality assumption by applying the JB and RJB tests.

Response: Thank for the suggestion! I have acknowledged the same on pg# 32, line# 463-64.

Reviewer # 2 Comments: 

I am thankful to you for providing me the valuable comments for the value addition to my work. 

1. More details on the simulations should be disclosed. For example, which software do you use? Additionally, in order to facilitate the reproduction of the study you should provide data. If you generate your distributions with Python, R, Matlab, etc, you could include the code as a supplementary material or upload at a repository such as Github.

Response: Thanks for providing guidance regarding the repository like Github. I have uploaded the codes required to generate the data for simulations and code for the replication of the study as well. The URL of the file is included in the revised cover letter. The software information is added to the manuscript at pg# 16, line# 275. 

2. Probably you could complement your results using a classification or a clustering algorithm. In other words, which features of the distribution influences the loss function the most? Do some tests perform better at detecting non normality against a Beta rather than against a Laplace distribution? You could use Self Organizing Maps or K-means or many other techniques in order to produce such representation. In this way instead of a ranking such as in tables 5,6,7, you could produce a ranking that informs on the different dimensions that affect the normality detection.

Response: Thanks again for the valuable comment, it helped me to improve my analysis of results. I have highlighted the top-five damaging distributions for each selected normality test at all sample sizes in table 8. This would help researchers to choose the best normality test based on the data distribution in hand. 

Again, thank you for giving me the opportunity to strengthen my manuscript with your valuable comments and queries. I have worked hard to incorporate your feedback and hope that these revisions persuade you to accept my submission.

Your Sincerely, 

Tanweer Ul Islam, 

National University of Sciences & Technology Islamabad, Pakistan, 44000.

Email: tanweer@s3h.nust.edu.pk ; Tel: +92-51-9085 3567

---

## [Decision Letter · Decision Letter 1]

9 Jul 2021

Min-Max Approach for Comparison of Univariate Normality Tests

PONE-D-21-14331R1

Dear Dr. Islam,

We’re pleased to inform you that your manuscript has been judged scientifically suitable for publication and will be formally accepted for publication once it meets all outstanding technical requirements.

Kind regards,

Saeed Mian Qaisar, Ph.D.

Academic Editor

PLOS ONE

Additional Editor Comments (optional):

I am pleased to tell you that your work has now been accepted for publication in the PLOS ONE Journal.

Thank you for submitting your work to this journal.

Reviewers' comments:

Reviewer's Responses to Questions

**Comments to the Author**

1. If the authors have adequately addressed your comments raised in a previous round of review and you feel that this manuscript is now acceptable for publication, you may indicate that here to bypass the “Comments to the Author” section, enter your conflict of interest statement in the “Confidential to Editor” section, and submit your "Accept" recommendation.

Reviewer #1: All comments have been addressed

Reviewer #2: All comments have been addressed

2. Is the manuscript technically sound, and do the data support the conclusions?

Reviewer #1: Yes

Reviewer #2: Yes

3. Has the statistical analysis been performed appropriately and rigorously? 

Reviewer #1: Yes

Reviewer #2: Yes

4. Have the authors made all data underlying the findings in their manuscript fully available?

Reviewer #1: Yes

Reviewer #2: Yes

5. Is the manuscript presented in an intelligible fashion and written in standard English?

Reviewer #1: Yes

Reviewer #2: Yes

6. Review Comments to the Author

Reviewer #1: All of the comments have been addressed by the author. My recommendation is "accept" without further comments.

Reviewer #2: The author correctly addressed all the previous comments of this reviewer.

Just please check the URL of github you provided, as I could not reach it, and include it in a footnote in the paper.

7. PLOS authors have the option to publish the peer review history of their article (what does this mean?). If published, this will include your full peer review and any attached files.

Reviewer #1: **Yes: **Asad ul Islam KHAN

Reviewer #2: No

---

## [Editor Report · Acceptance letter]

21 Jul 2021

PONE-D-21-14331R1 

Min-max approach for comparison of univariate normality tests 

Dear Dr. Islam:

I'm pleased to inform you that your manuscript has been deemed suitable for publication in PLOS ONE. Congratulations! Your manuscript is now with our production department. 

Kind regards, 

on behalf of

Dr. Saeed Mian Qaisar 

Academic Editor

PLOS ONE